# Systematic Review and Meta-Analysis on the Effects of Astaxanthin on Human Skin Ageing

**DOI:** 10.3390/nu13092917

**Published:** 2021-08-24

**Authors:** Xiangyu Zhou, Qingming Cao, Caroline Orfila, Jian Zhao, Lin Zhang

**Affiliations:** 1The Key Laboratory for Special Medical Food Process in Hunan Province, Central South University of Forestry and Technology, Changsha 410004, China; 605460940tiffany@gmail.com (X.Z.); zhanglin840514@126.com (L.Z.); 2Faculty of Environment, School of Food Science and Nutrition, University of Leeds, Leeds LS2 9JT, UK; C.Orfila@leeds.ac.uk; 3School of Chemical Engineering, The University of New South Wales, Sydney, NSW 2052, Australia; jian.zhao@unsw.edu.au

**Keywords:** astaxanthin, skin ageing, antioxidant, anti-inflammation

## Abstract

Context: Astaxanthin (ASX), a xanthophyll carotenoid derived from microalgae *Haematococcus pluvialis*, mitigating skin photoaging and age-related skin diseases by its antioxidant and anti-inflammatory effects in animal studies. Objective: The aim was to systematically evaluate if ASX applications have anti-ageing effects in humans. Methods: A comprehensive search of PubMed, Scopus and Web of Science found a total of eleven studies. Nine randomised, controlled human studies assessed oral ASX effects and two open-label, prospective studies evaluated topical, oral-topical ASX effects on skin ageing. *GetData Graph Digitizer* was used to extract mean values and standard deviations of baseline and endpoint, and Cochrane Collaboration’s tool assessed RoB for all included studies. Review Manager 5.4 was used to conduct meta-analysis of RCTs; the results were reported as effect size ± 95% confidence interval. Results: Oral ASX supplementation significantly restored moisture content (SMD = 0.53; 95% CI = 0.05, 1.01; I^2^ = 52%; *p* = 0.03) and improved elasticity (SMD = 0.77; 95% CI = 0.19, 1.35; I^2^ = 75%; *p* = 0.009) but did not significantly decrease wrinkle depth (SMD = −0.26; 95% CI = −0.58, 0.06; I^2^ = 0%; *p* = 0.11) compared to placebo. Open-label, prospective studies suggested slightly protective effects of topical and oral-topical ASX applications on skin ageing. Conclusions: Ingestion and/or topical usages of ASX may be effective in reducing skin ageing and have promising cosmetical potential, as it improves moisture content and elasticity and reduces wrinkles.

## 1. Introduction

Astaxanthin (ASX) is a xanthophyll carotenoid, chemically known as 3,3′-dihydroxy-β,β-carotene-4,4′-dione. It was originally isolated from lobster by Kuhn in 1938 [1]. Its characteristic bright red-orange colour occurs in salmon flesh, thus it is widely used as a colourant in aquaculture feeds [2]. ASX is primarily biosynthesised by various algae, bacteria and fungi and is consumed by marine animals, such as salmon, trout, crab, lobster and shrimp [3]. The chlorophyte alga, *Haematococcus pluvialis*, is reported to accumulate the highest levels of ASX in nature [4]. ASX occurs predominantly as monoesters or diesters that may be hydrolysed in the small intestine, which facilitates its absorption and transportation into plasma and erythrocytes [5] and accumulation in the skin effectively [6], making human skin the major physiological target. Skin ageing is becoming a global challenge due to the rapidly increasing human lifespan around the world and intensive ultra-violet rays contributing to the destructions of ozone layers [7]. This finding attracts a great deal of scientific interest to investigate if ASX could prevent or even slow down skin ageing, and have a promising cosmeceutical potential.

The skin comprises epidermis, dermis and hypodermis, which form the protectively outermost barrier against external environmental stresses such as repeated sun UV ray exposure, microorganisms or pathogens invasion, physicochemical agents and excessive transpiration of internal moisture [8]. Photo ageing accounts for most age-related changes in skin appearance, which is caused by the superpositioning of chronic UV-induced deteriorations on the intrinsic ageing process [9]. Aged skin is characterized by skin dryness, deep wrinkling, laxity and increasing transepidermal water loss (TEWL) (the loss of water passing from the inside body through the epidermis) and epidermal barrier dysfunctions [10].

UV radiation comprises three types: UVA, UVB and UVC. UVC is filtered out by atmospheric ozone for the most part, while both UVA and UVB can cause a biological change in the skin [6]. UVA, which contributes up to 95% of the total UV exposure, results in skin photoaging by penetrating skin dermis and degrading the epidermal layer supporters: dermal collagen and elastin which are produced by the fibroblasts and are responsible for skin strength and elasticity, respectively [11]. UVB mainly affects the epidermis and epidermal keratinocytes that could communicate with dermal fibroblasts to facilitate relief to exogenous and endogenous damages [8]. UVB could also damage cellular macromolecules (nucleic acids, lipid and protein) and induce generations of high levels of ROS, stimulating chronic inflammation (See Figure 1) [6]. Moreover, intrinsic skin ageing rapidly accelerates especially after the age of 30 [12], because cell metabolic activities, such as the capacity of endogenous antioxidants enzymes (i.e., glutathione peroxide (GPx) from epidermis) for the elimination of the oxidized biomolecules, are reduced as with age, promoting an accumulation of intracellular reactive oxygen species (ROS) and inflammatory responses [13]. Meanwhile, the gradually disrupted epidermal barrier accelerates skin ageing [8] owing to the loss of functions of several important factors, such as proteases, proteases inhibitors, corneocytes (consist of dead epidermal cells) and lipids [8].

ASX is structurally similar to β-carotene (See Figure 2) but has 40 times stronger antioxidant activity, as its polar ionone rings on both ends can quench free radicals and other reactive oxygen species (ROS), and the thirteen conjugated double, polyunsaturated bonds can remove high-energy electrons [14]. Its amphipathic structure with polar-nonpolar-polar characteristics allows ASX to be inserted into the bilayers of cell membranes, confines lipoperoxidation promoters to penetrate across the lipid bilayer and thus reduces peroxidation-caused damages [15]. Dietary ASX supplementation could lower wrinkle formations, decreased TEWL and maintain epidermal barrier functions significantly in the dorsal skin of HR-1 hairless mice under UVA exposure by its antioxidative effects, as compared to the control group [6]. In addition, ASX also has potential anti-inflammation effects by inhibiting inflammatory mediators [16]. Atopic dermatitis (AD) is a disease that is commonly found in aged people with reduced skin barrier functions due to slowing metabolic activity [17]. ASX cream could improve the development of phthalic anhydride (PA)-induced AD in HR-1 hairless mice by inhibiting the releases of various inflammatory cytokines [18]. However, the beneficial effects found in animal studies cannot be directly translated into protective effects in humans partially due to the different bioavailability of ASX across species. This review aims to conduct a systemic review and meta-analysis to evaluate the effects of ASX on human skin ageing, based on currently published trials and highlight future research directions.

## 2. Method

This review was performed according to the Preferred Reporting Items for Systemic Reviews and Meta-Analyses (PRISMA) guidelines, including identification, screening, eligibility and inclusion [2].

### 2.1. Literature Search

To conduct a comprehensive systematic literature search, the controlled vocabulary and free text terms and Boolean operators “AND” and “OR” were used. The following text words: (“astaxanthine” OR “astaxanthine” OR “astaxanthin” OR “astaxanthins” OR “ASX” OR “carotenoid”) AND (“skin” OR “skin derm *”) appeared in all fields of the articles. Multiple databases, including PubMed, Scopus, and Web of Science, were searched, yielding articles published in English between 2001 and 2021.

### 2.2. Inclusion and Exclusion Criteria

A study was included if it met all of the following inclusion criteria: (1) published paper had full length and was in a peer-reviewed source (2) it was conducted in human subjects; (3) it evaluated skin ageing, including skin parameters and biomarkers related to mechanisms; (4) the investigational product must contain ASX; (5) it involved either oral, topical or combinational ASX applications; (6) it provided available qualitative and/or quantitative data on skin ageing before and after treatments. A study was excluded if it met one or more of the following exclusion criteria: (1) published paper was in a non-peer-reviewed source (i.e., website, magazines); (2) it was in the abstract form; (3) it was an animal study or a cell line study; (4) the investigational product contained additional bioactive agents which exerted the major benefits rather than ASX or did not contain ASX; (5) it was secondary research (i.e., narrative review, systemic review, meta-analysis, notes and conferences etc.); (6) it investigated diseases which were not related to aged skin; (7) it was a duplicate publication. Although the secondary research papers were excluded, the relevant articles in bibliographies of these publications were manually identified for additional information.

### 2.3. Data Extraction

The following data were extracted: study design, study populations (age (years), gender, country and health conditions), the number of individuals in each group (intervention or control), study duration (weeks), content and dosage (mg/d) of each group, a form of applications, conditions of test rooms (°C room temperature (R.T.); % relative humidity (R.H.), seasons during study periods, UV-applications, measurement instruments, and skin parameters (measuring sites) with units were transcribed into a table.

Other captured data, including the baseline and endpoint measure (mean value) and its variability (either standard deviation or standard error of the mean), were extracted by applying GetData Graph Digitizer as most publications presented results into figures [19]. These values were transcribed into an Excel spreadsheet with defined units for the outcomes (i.e., wrinkle depth was presented in µm). The following outcomes were captured: moisture content, sebum content, skin elasticity, TEWL, wrinkle depth, sebum oil, presence of oxidative stress biomarkers (thymine dimers and 8-hydroxy-2′-deoxyguanosine (8-OHdG), malondialdehyde (MDA)), mRNA expressions of procollagen type I and fibrillin-1, Matrix Metalloproteinase (MMP)-1, and MMP-2, interleukin (IL)-1α, minimal erythema dose (MED), mean depth of texture, the total area of corneocyte, inflammatory biomarkers (C-reactive protein (CRP)), lipid droplet size and corneocytes desquamation. As some of the selected studies performed outcome measurements at several key points during study periods, the effects of the longest durations (endpoints) were extracted.

### 2.4. Statistical Analysis

The randomised, controlled studies (RCTs) were grouped according to the similarity of each outcome related to skin ageing, and meta-analysis was conducted for each group independently using Review Manager 5.4 [20].

Statistical analysis used the random-effect model to calculate the mean difference and standard deviations (SDs) for continuous variables. Mean difference was determined by the value at the endpoint subtracted from the value at baseline. If RCTs only presented the change rate (post-intervention/pre-intervention), then mean values for the pre-intervention can be hypothesized referring to results of other studies in this review carrying similar outcome measures [21]. If for the RCTs, the SDs were reported only for the baseline and final point in both intervention and placebo groups, then SDs for changes from baseline were calculated by the formula:$${\mathrm{SD}}_{\mathrm{change}}=\sqrt{{\mathrm{SD}}_{\mathrm{baseline}}^{2}+{\mathrm{SD}}_{\mathrm{final}}^{2}-\left(2\times \mathrm{Corr}\times {\mathrm{SD}}_{\mathrm{baseline}}\times {\mathrm{SD}}_{\mathrm{final}}\right)}$$
assuming a correlation coefficient (Corr) = 0.8 [21]. If in RCTs, the SDs were reported for neither baseline nor final point, then change-from-baseline SDs were imputed according to other studies in this review which used similar outcome measures [21]. In terms of cases reported as the standard error of the mean (SEM), SD was calculated by SD = SEM × $\sqrt{n}$ (n is the number of subjects) [21].

Moreover, when an identified study showed multiple comparisons (groups/treatment arms) such as low dose and high dose group, each comparison between ASX and placebo was considered as a separate trial. The sample size of the control group was divided evenly among the comparisons to avoid double counting the subjects and consequent unit-of-analysis error [22].

If RCTs compared the effects of rosehip powder to ASX, then the sample size of the ASX group was regarded as the intervention group and the rosehip powder group was the placebo group. As rosehip powder did not contain ASX, it was reasonable to assume it was a placebo group.

Although the included RCTs have assessed several skin ageing parameters, meta-analysis was conducted only if there were at least 3 available identified studies for each parameter.

Some of the RCTs measured skin elasticity according to gross elasticity (R2), net elasticity (R5), biological elasticity (R5) and analysed wrinkles based on several parameters: deepest point of the deepest wrinkle, mean depth of the deepest wrinkle, the maximum width of the deepest wrinkle, the area ratio of all wrinkles, mean depth of all wrinkles and volume ratio of all wrinkles. This meta-analysis extracted and analysed values of gross elasticity (R2) and mean depth of all wrinkles because they suggested the whole changes in skin elasticity and wrinkles during study periods better than other parameters. As Higgins I ($I^{2}$) represents the level of heterogeneity among the included studies and measures the proportion of inconsistency that cannot be explained by chance alone [23], ranging from 0% to 100%. $I^{2}$ < 25% indicated very low, 25% < $I^{2}$ < 50% reported low heterogeneity, 50% < $I^{2}$ < 75% indicated moderate heterogeneity, whereas $I^{2}$ ≥ 75% corresponded to substantial-high heterogeneity [24]. If there was a small or absent heterogeneity, a fixed-effects model was used in the meta-analysis [24]. As included studies did not apply the same outcome measure and scales, the standardized mean difference was proposed to be used [24].

The probability (*p*) value of ≤0.05 was regarded as statistically significant. The result of the overall effect test was shown as a standardized mean difference and 95% confidence interval (CI). The small effect sizes were 0.2 and below, the medium size ranged from 0.3 to 0.7 and the large effect size were 0.8 and above.

The methodological quality of included studies was systematically assessed using Cochrane Collaboration’s tool for evaluating the risk of bias (RoB) [25], which involves seven items, including random sequence generation, allocation concealment, blinding of participants and personnel, blinding of outcome assessment, incomplete outcome data, selective reporting and other bias. Following the guidelines in the Cochrane handbook, there are three options for judgments: low risk of bias in green, unclear risk of bias in yellow, and high risk of bias in red, respectively [26].

Furthermore, the results for topical, oral-topical ASX administrations on skin ageing from open-label, prospective studies, and extracted skin parameters from RCTs which were not pooled into meta-analyses, would be summarized in percentage, calculated by: Improvement from baseline = ((endpoint mean − baseline mean) ÷ baseline mean) × 100%.

## 3. Results

### 3.1. Literature Search

As shown in Figure 3, the literature search identified 3525 records. Of the 3525 records, 1516 duplicates were excluded. By retrieving the strength of 2009 records, 824 secondary papers and 1071 non-human studies were excluded. The remaining 114 were considered potentially relevant, and their full-length versions were reviewed. A total of 103 papers were excluded for reasons shown in Figure 2 (corresponding to inclusion and exclusion criteria). Thus, 11 of 114 full-text publications were included.

### 3.2. Overview of All Studies

The key characteristics of the 11 clinical studies are summarized in Table 1. Eight randomised controlled studies (RCTs) were pooled into a meta-analysis to assess the effects of oral ASX capsules on skin ageing, and one RCT provided additional information about the antioxidants and anti-inflammatory properties of ASX [27]. There were three open-label, prospective trials [13,28,29] to evaluate topical, oral-topical applications of ASX on skin ageing. Tominaga et al. (2012) [29] and Seki et al. (2001) [30] conducted two clinical trials in the same article. One open-label, prospective study assessed the effect of oral ASX capsules intake on influential factors of epidermal barrier functions [31].

Among eight RCTs pooled into a meta-analysis, there were six randomised, double-blinded and placebo-controlled studies [9,13,28,29,32,33]. Three studies used canola oil [13,28,29], one study used medium-chain triglycerides [9], one study used a filling agent [33] and one study used hip-rose powder as a placebo group [32]. Two randomised, single-blind and placebo-controlled used canola oil [34] and rosehip powder [35] as placebo, respectively. Those studies applied ASX which was derived from *Haematococcus pluvialis* microalgae. RCTs commonly used algae extract based on various oil formulations (i.e., canola oil, safflower oil, rice bran oil and olive oil) as the intervention capsules, while Ito et al. (2018) [33] applied dispersant technology and supercritical CO_2_ extraction technology to achieve solvent-free algal ASX. Two studies investigated the additive and synergistic effects of a low dose of ASX (2 mg/d) and other antioxidants, such as collagen hydrolysate [9] tocotrienol [28], on skin ageing. Other studies applied 4 mg/d to 12 mg/d of oral ASX capsules, and the duration of RCTs ranged from 4 weeks to 16 weeks. The RCTs were conducted in four different countries: Japan, Thailand, South Korea, and the United States. The number of participants ranged from 16 to 65, and they were in the middle-aged age range with a mean age of 40 to 50 years, with defined age-related signs such as wrinkle and dryness at baseline. Furthermore, 75% of the studies were conditioned to mean values of 20 °C relative room temperature and 55% relative humidity, and 62.5% of studies emphasized their trials starting from autumn to winter. Only Ito et al. (2018) [33] specialized on the efficacy of ASX on the UV-irradiated skin deterioration by measuring MED, which is the amount of UV radiation producing minimal erythema on individual skin, moisture maintenance and TEWL at irradiated skin on healthy subjects (the used UV rays equivalent to exposure to the sun for 1.5 h in the Japanese summer). Yoon et al. (2014) [9] only applied UV irradiation on skin biopsy samples taken from buttock area in a small group of subjects, and evaluated the UV-induced oxidative and inflammatory biomarkers rather than skin parameters. Skin conditions of participants in other studies were affected by individual ASX pharmacokinetics, intrinsic ageing and environmental conditions, such as humidity, temperature and daytime UV exposure time depending on specific geographic positions and seasons (test period). Most of the studies conducted skin measurements at cheek. Some studies assessed dermatological parameters at other measuring sites including outer canthus, forehead, and UV-irradiated sites. The majority applied different dermatological measurements instruments on the same skin parameters.

As for the open-label, prospective studies, the first study by Seki et al. (2001) [30] was the skin repeated application test which assessed the effects of ASX cream combined with other cosmetics on human skin using 11 subjects with a mean age of 35 years. The second study by Seki et al. (2001) [30] was a preliminary study on the beauty effect of ASX cream combined with other cosmetics using three subjects with normal skin, dried skin and mixed skin type, respectively, at a mean age of 33 years. Study 1 by Tominaga et al. (2012) [29] evaluated the effects of the combination of both oral supplementation and topical administration of ASX using 30 females (10 with dried skin).

### 3.3. Meta-Analysis for Oral ASX Effects

The eight RCTs involved 162 subjects in the ASX supplement group and 131 subjects in the placebo group. They were initially divided into three groups according to the measured outcomes, including moisture content, skin elasticity and wrinkle depth and then subjected to the meta-analysis.

Figure 4A displays the forest plot of the meta-analysis of five studies with respect to combined moisture content estimates comparing the placebo group with the group of participants orally supplemented with ASX. For this outcome, supplementation resulted in statistically significant improvement (Z = 2.17, *p* = 0.03), as based on the overall effect size of 0.53 (95% CI = 0.05, 1.01).

In the meta-analysis with 6 trials (Figure 4B) that evaluated skin elasticity, oral ASX supplementation also significantly improved the outcome (Z = 2.61, *p* = 0.009) in comparison to the placebo group, with an overall effect size of 0.77 (95% CI = 0.19, 1.35).

No significant differences (Z = 1.58, *p* = 0.11) were identified in studies between the ASX treatment and placebo groups in studies evaluating the wrinkle depth. Figure 4C shows a general effect size of −0.26 (95% CI = −0.58, 0.06).

For other extracted skin parameters, sebum content and sebum oil, which was reported in a single study, as well as TEWL which were reported by four trials without enough data, were not pooled into meta-analysis.

### 3.4. Oral ASX Effects on Other Skin Parameters Based on RCTs

TEWL was reduced from baseline by 49.52% after administrating with 2 mg/d of ASX and 3 mg/d of collagen hydrolysate for 12 weeks while there was a 36.96% of TEWL reduction in the control group [9]. It is consistent with another study that six weeks of 6 mg/d of ASX supplementation resulted in a significant decrease in TEWL from the baseline while there was an increase in TEWL in the control group (study 2) [29].

Sebum content of the cheek has been maintained by four weeks of oral ASX and tocotrienol intake from baseline, while sebum content was decreased by 7.21 (without unit) in the placebo group at the end of the study [28]. A substantial reduction in sebum oil of cheek from baseline was observed after six weeks of applications of 6 mg/d of ASX oral supplementation compared to the placebo group [29].

### 3.5. Open-Label, Prospective Studies

Two open-label prospective studies measured the effects of topical and oral-topical ASX applications on skin ageing.

The moisture content of the left outer canthus was significantly increased from baseline by 3.32% after three week-topical administration of 280 mg/d of ASX cream (study 1) [30]. The improvements in skin conditions such as dryness, flushing, itching and inconsistency with makeup were observed from questionnaire responses after three weeks of repeated ASX topical applications (study 1) [30]. The second study by Seki et al. (2001) [30] also showed a 106.67% elevation in skin moisture content of left outer canthus after two-week application of 140 mg/d of ASX-based cream in participants with dried and mixed skin. An antiwrinkle effect of ASX under eyes and outer canthus was observed in three subjects with different skin types by using magnified photographs. Skin conditions including moistness, smoothness, and elasticity obtained from skin-palpation and skin inspection by beauty specialists, were improved in all three subjects.

Significant wrinkle reductions and elasticity improvements were observed at outer canthus by 2.27% and 3.39%, respectively, after eight weeks of applications with 6 mg/d of oral ASX capsules and 94.18 mg/d of topical ASX cream [29]. A combination of ASX techniques showed a significant increase in moisture content of the corneocyte layer, the total area of the corneocyte, and the mean depth of skin texture in the cheek in 10 participants with dried skin [29].

### 3.6. Risk of Bias

The Cochrane Collaboration’s tool was applied to assess the quality of RoB for this systemic review. Figure 5 shows the assessment of RoB at the domain level revealed an unclear RoB for most studies.

Open-label, prospective study is regarded as being subjected to a high bias because it is limited by the impact of knowledge of nonrandomized treatment allocation and unblinding. The behaviour of the investigator or subject would be potentially impacted when they were assigned to a specific trial arm due to psychological effects [36], which can introduce a high RoB to the random sequence generation, allocation concealment, blinding of participants and personnel and blinding of outcome assessment. For example, the participant was required to take only one known substance, which was much easier for the investigator to explain the procedure to the subjects than complicated RCTs [37]. This would also achieve higher compliance and better outcomes [37]. Because subjects may research the potential outcomes and the impact of ASX, their endpoint at reporting behaviour of exact effects would be compromised [37], leading to unclear to high RoB at reporting bias.

Moreover, randomized double-blinded and single-blinded controlled studies selected in this review were confined by poorly reported study design, leading to the unclear RoB at nearly all domain levels. Further, albeit SDs for changes from baseline in RCTs were not reported in some studies, the attrition and reporting bias was reduced to low levels in several RCTs [9,32,35] by prespecified statistical analysis and formula (See Section 2.4).

Besides, some of the selected studies were either conducted by various study investigators affiliated with cooperative institutes or commercial entities or funded by commercial entities, with a potential financial conflict of interests [9,13,31,33]. However, others did not declare any potential conflicts of interests [27,28,29,30,32,34,35]. That makes it difficult to access the exact influences of commercial interests on the analysis, contributing to an unclear risk of potential bias.

## 4. Discussion

### 4.1. Oral ASX Effects

Based on the pooled results of meta-analysis about skin moisture contents and skin elasticity betterments, and positive results of TEWL, sebum content and sebum oil from RCTs, this systemic review could partially agree with previous narrative reviews, indicating the ASX oral applications probably prevented skin ageing in healthy middle-aged participants with defined age-related signs at baseline [2,14,38].

However, due to absent statistical heterogeneity (I^2^ = 0, Figure 4C), it is reliable and robust to consider that within each type of intervention, oral ASX supplementation, compared with control group, could not significantly decrease wrinkle depth (SMD = −0.26; 95% CI = −0.58, 0.06; *p* = 0.11) (Figure 4C). That may be explained by small involvements of participants in each study because it is difficult to distinguish whether the result of the small true-effect size was due to a real biological action of ASX or random variation by chance [39,40].

Moreover, because of the relatively high heterogeneity in moisture content (I^2^ = 52%) and skin elasticity (I^2^ = 75%) among studies, it is less sure that ASX intervention could consistently have beneficial effect sizes of 0.53 and 0.77 on moisture content and skin elasticity, respectively. Such heterogeneity might mainly be because of synergetic effects of other bioactive molecules, different measuring sites, ASX formulations, or environmental factors in the evaluated area during the study periods.

Participants in the study by Yamashita et al. (2002) [28] were supplemented with low a dose (2 mg/d) of ASX and 40 mg/d of tocotrienol for a short term (four weeks), with the best effect size of 2.11 (95% CI = 0.81, 3.40) as compared to other studies which applied higher dosage of ASX (4 mg/d) and the study by Yoon et al. (2014) [9] which used 2 mg/d ASX and 3 g/d collagen hydrolysate for a relatively long time. The collagen hydrolysate has a relatively low antioxidative activity as oxidative biomarkers, thymine dimers and 8-OHdG levels did not significantly differ between groups after UV-irradiation [9]. However, the tocotrienol has antioxidative action 40 to 60 times stronger than tocopherol by scavenging peroxyl radicals [41] and ASX is 100 to 1000 times higher than tocopherol via quenching singlet oxygen [28]. These combinations could exert a synergistic action due to different modes of action. Although tocotrienol was found in rice bran oil which was used by Phetcharat et al. (2015) [32] in ASX oil formulations, the higher dosage of tocotrienol in Yamashita et al. (2002) [28] might contribute to the better effects. Both might be responsible for the moderate heterogeneity within this group. Moreover, the study by Ito et al. (2018) [33] administrated to subjects 4 mg/d of ASX obtained by supercritical CO_2_ extraction technology, which achieved an improved absorbability of ASX in the subjects. They achieved a relatively high effect size of 0.62 (95% CI = −0.24, 1.48) on restoring UV-induced decrease of moisture compared to similar studies using oil formulations of ASX. Thus, it appears that the supercritical CO_2_ extraction technology might strengthen the efficacy of ASX and could be assumed as source of heterogeneity. Therefore, future studies might be recommended to apply these technologies to process ASX which could magnify the effect of ASX in humans for more sound quantitative values. The effect size of −0.22 (95% CI = −0.07, 0.67) in the Phetcharat et al. (2015) [32] study might be attributable to the placebo group, as rosehip powder had similar protective effects to restore skin moisture as ASX, contributing to the moderate heterogeneity.

The studies by Tominaga et al. (2017) [13], Phetcharat et al. (2015) [32] and Yoon et al. (2014) [9] measured skin elasticity at the cheek resulting in a relatively smaller effect size than studies by Yamashita (2005) [34] and Tominaga et al. (2012) [29] which measured skin elasticity at left outer canthus. Because several factors such as muscular asymmetry, dietary pattern, lifestyle habits including sleeping positions, uneven sun exposure time and frequency to apply sunscreen, might result in different skin deterioration rates between the cheek and left outer canthus, thus contributing to the high heterogeneity [42]. Moreover, the study by Tominaga et al. (2017) [13] was conducted from August to December, a period in the experimental sites during which exposure to the strongest UV radiation during the summer was followed by the gradually declining air humidity and temperature during the autumn and winter months. These changing environmental factors resulted in significant degradation of skin elasticity, so that even high dosage (6 mg/d and 12 mg/d) of ASX capsules exerted a very small and similar effect size of 0.04, without a dose-response effect. This is because strong UV radiation could upregulate the activity of skin fibroblast-derived elastase to impair the elastic fibre configuration with subsequent loss of skin elasticity [43]. Moreover, one study reported that after acclimation to low humidity (40% relative humidity (RH)) for 30 min compared with high humidity (70% RH), there was a decrease in skin elasticity in the eyelids of participants [44]. Low temperature exaggerated a decrease in skin barrier function and skin elasticity and increased susceptibility of skin towards environmental stresses [45]. Studies by Yoon et al. (2014) [9] and Phetcharat et al. (2015) [32] measured skin elasticity at the cheek and were conducted in different countries for a relatively shorter term but did not clarify starting time of the study. Thus, the environmental damages on the skin during the study period were unknown, which might lead to high heterogeneity and low reliability for the relatively high efficacy of ASX in those two studies. Thus, future studies should elucidate the environmental conditions (humidity, temperature, and UV radiation) during the study period and apply various doses of oral ASX to investigate if there is a dose-response effect in humans.

### 4.2. Topical and Oral-Topical ASX Effects

The included open-label, prospective studies might only suggest that topical, and oral-topical ASX applications could slightly prevent human skin ageing because of no separate control groups included, while the pre-treatment conditions were used as an internal control for each study participant.

Furthermore, based on the very short intervention time of studies (two or three weeks) in Seki et al. (2001) [30], the reduction of wrinkles and increase of elasticity might be related to the increasing skin moisture content, but not necessarily the modification of collagen structure by ASX. Because of the usage of other cosmetics with ASX in two trials by Seki et al. (2001) [30], the mechanism of actions of ASX and its beautifying effects need further studies to confirm. Moreover, the positive results from very small numbers of participants measured by subjective questionnaires, inspection and dermatological instruments were likely to be unreliable (Seki et al., 2001) [30]. The trial by Tominaga et al. (2012) [29] applied topical ASX cream without any other effective base materials on participants for a longer time and measured skin parameters using objective instruments, which might lead to more reliable positive results. Therefore, it is relatively certain that synergistic effects of combination techniques could lead to much more benefits for skin health and oral supplementation might have more sustained and pronounced effects than topical application [29,46]. This is because oral ASX intake can convey signals to the body to increase the production of collagen while topical ASX usage on healthy skin is usually unable to penetrate the epidermal barrier deep enough into the underlying tissues and structures to be transferred to the systemic circulation to produce collagen on a cellular level [47]. In addition, the topical application tends to produce local impacts at the skin epidermal layer temporarily and lose the effects quickly [47].

Therefore, it is urgent to conduct more randomised, controlled trials on larger sample sizes for a longer time and use objective dermatological instruments to achieve more reliable results about the effect of topical and combinational ASX applications on skin health.

### 4.3. Mechanisms

Although the exact underlying mechanism involved in the prevention of human skin ageing by ASX was not clear, this might be a result of its antioxidant and anti-inflammatory properties and its beneficial effects on skin barrier integrity improvements.

Excessive accumulations of reactive oxygen species (ROS) from repeated UV exposures on epidermis and dermis, and biologically unbalanced redox state due to intrinsic ageing, could increase releases of inflammatory cytokines such as interleukin (IL)-1, IL-6, IL-8 and inflammation-related tumour necrosis factor-α (TNF-α). These, in turn, could stimulate epidermal keratinocytes and dermal fibroblasts in an autocrine manner, then upregulate the mRNA expressions, proteins and enzymatic activity of MMPs, including MMP-1, MMP-3, and MMP-9 [13]. Meanwhile, the estrogen levels and producing rate begin to decline in women after the age of 30, which promotes the accumulation of MMPs [48]. MMPs could promote the degradation of collagen and elastic fibres by increasing elastase activity, leading to the formation of wrinkles and loss of skin elasticity [49], while elevated epidermal IL-1α levels could induce productions of other pro-inflammatory cytokines, such as IL-6 and IL-8, and then promote skin dryness [13].

ASX can scavenge singlet oxygen radicals in the epidermis and dermis, leading to a restoration of skin elasticity and reduction in wrinkle depth [50]. ASX could also upregulate endogenous antioxidative enzymes, including superoxide dismutase 2 (SOD 2), catalase (CAT), and glutathione peroxidase 1 (GPX1) to suppress the activation of ROS-producing enzymes and xanthine oxidase (XO) in UV-irradiated cells [51]. Furthermore, ASX could prevent the production of lipid peroxides, whereby facilitate maintaining the declined sebum content with age [28,31] and preventing sebum lipids from peroxidation, and decreasing rough skin and ageing odour [29]. 2 mg/d of dietary ASX supplementation for 8 weeks was found to decrease DNA oxidative damage biomarker (plasma 8-OHdG) by 35.29% and lower the concentrations of inflammatory markers (plasma CRP) by 33.56% in young healthy females, as compared to placebo groups [27]. Treatments with ASX reduced UV-induced increases of IL-1β and TNF-α in HaCaT keratinocytes [52]. The production of MMP-1 by fibroblasts which were cultured in UVB-irradiated keratinocytes medium was dose-dependently suppressed by ASX treatment (0, 1, 5 or 10 µM of ASX) [13]. Furthermore, oral dosing with ASX and collagen hydrolysate could increase procollagen type I mRNA by 240%, decrease MMP-1 (collagenase) mRNA by 68%, suppress MMP-12 (elastases) mRNA by 77% and increase fibrillin-1 mRNA by 87.2% in human photoaged facial skin, compared with those expressions in the placebo group after 12 weeks of treatment [9].

Moreover, a study by Chalyk et al. (2017) [31] showed the beneficial effects of ASX on maintaining the epidermal barrier functions and integrity by inhibiting the pronounced and excessive corneocyte desquamation significantly and decreasing the plasma oxidation-related MDA concentration by 21.7% significantly in the older participants. Additionally, a study by Komatsu et al. (2017) [6] showed that ASX suppressed the protease inhibitor expressions in mice epidermis, thus stimulating proteolysis of the filaggrin which facilitated the production of natural moisturizing factors, resulting in decreasing desquamations of terminally differentiated corneocytes. ASX could significantly improve TEWL, normalize corneocyte conditions and protect the keratinocyte differentiation and cornification from oxidative damages and inflammations in the epidermis [29].

### 4.4. Limitations and Recommendations for Future Work

Several limitations should be considered and acknowledged for the slightly positive results. Firstly, the main limitation is only a small number of included clinical trials (less than 10 studies) pooled into a meta-analysis, and the number of subjects in each study was small (less than 100); therefore, the findings in this review should only be considered as preliminary. This also prevented subgroup analysis for identifying the sources of heterogeneity, assessing the treatment effectiveness [53] and using funnel plots for assessing the potential role of reporting bias [54]. Because when there are fewer than 10 studies, the power of subgroup analysis and the testes for funnel plot asymmetry is too low to distinguish chance from real differences and substantial reporting bias [53,54].

Moreover, there is a lack of randomised controlled trials for topical and oral-topical ASX applications on skin ageing, preventing making a robust conclusion. Secondly, the method of data extraction (GetData Graph Digitizer) might limit the statistical power and accuracy. Moreover, data of baseline value and SDs of changes from baseline in the second study by [29] were referred to its first study [29], decreasing the reliability of meta-analysis for the wrinkle-depth group. Thirdly, the beneficial effects of ASX may depend on the bioavailability which is influenced by absorption, metabolism and disposition in skin tissues. ASX is a lipophilic dietary supplement. Human absorption of ASX ranges from 6% to 34% after 4 h and the eliminate half-life of ASX was reported to be from about 16 h to 30 h, which are responsible for a relatively low oral bioavailability [55,56]. Thus, the RCTs was usually administrated at least 3 mg/d of ASX with or after meal to enhance and optimize the bioavailability because of the presence of dietary lipids, which was 2.4 times higher bioavailability of ASX than before meal [57]. However, all included RCTs of meta-analysis did not measure the concentrations of ASX in the skin and elucidate whether these ASX levels were great enough to exert relevant biological activity. Next, most of the selected studies used different oil formulations of ASX as intervention capsules, which may contain other potentially beneficial ingredients for skin health, such as canola oil, which has omega-3 fatty acids for balanced hydration betterment [58]. The positive outcomes of ASX supplementation might be influenced by the formulation vehicles. Moreover, the quality assessment of all included studies was performed and most of them had a relatively unclear RoB, resulting in relatively poor study quality. The potential bias due to commercial interests in several studies might also contribute to the low reliability of current data. Positive outcomes of clinical trials could be affected by commercial interests because of several factors, such as applications of subjective dermatological assessments, selectively reporting outcomes and presenting results in a biased manner [59]. Finally, most of the selected trials were conducted on healthy Japanese females, potentially restricting the generalization of the results to a wider population demographics and assessment of whether there were sex-dependent responses. Moreover, the included studies did not restrain the use of cosmetics such as skin care products by participants. Meanwhile, it was unclear if included participants preferred to intake ASX rich foods like shrimp and salmon, and the effects of exact dietary ASX intake by participants were not estimated by the included studies. Both might obscure the real actions of oral ASX supplements.

It is thus necessary for future studies to design more large-scale randomized, blinded, controlled trials by recruiting more middle-aged healthy female and male participants across various regions especially for people from Western countries, using more objective dermatological assessments and reporting the original data. It is also desirable to apply supercritical CO_2_ extraction technology to generate a solvent-free and isolated form of ASX. Furthermore, it would help if participants were advised to refrain from using cosmetic products, especially skincare products during study periods and include the concentrations of ASX in the skin when measuring skin parameters.

## 5. Conclusions

Given the results of the meta-analysis and weak evidence from open-label, prospective studies, ASX oral and/or topical applications may delay and improve the signs of skin ageing by enhancing moisture content and skin elasticity, reducing facial wrinkles and sebum oil due to its antioxidant, anti-inflammatory effects and improved effects on skin barrier integrity. Oral supplementations might be more sustained and pronounced than topical applications. A synergistic skin protective effect was found in the combinational usages. However, the reliability and strength of the evidence were limited by small sample sizes, imperfect study design, method of data extraction and potential conflicts of interests. More large-scale and robust RCTs, as well as and usages of objective dermatological assessments are required to confirm the mechanisms.

## Figures and Tables

**Figure 1 nutrients-13-02917-f001:**
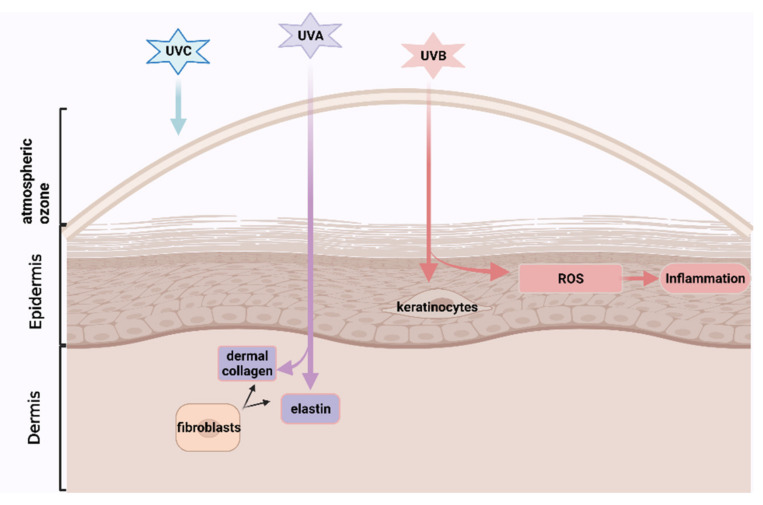
The mechanisms that UV-rays promote skin photoaging (created with BioRender.com. https://app.biorender.com/illustrations/61123dbdedaefd00a5e711d7. Accessed on 12 August 2021).

**Figure 2 nutrients-13-02917-f002:**
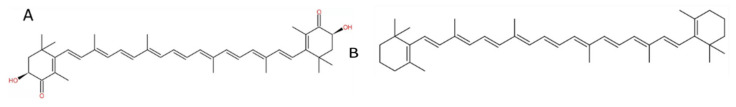
Chemical structure of ASX (**A**) and β-carotene (**B**) (created with KingDraw http://www.kingdraw.cn/en/index.html , accessed on 2 February 2021).

**Figure 3 nutrients-13-02917-f003:**
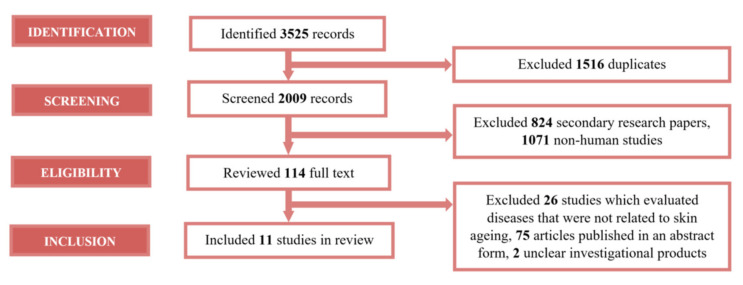
PRISMA flow diagram summarized identified studies during the literature review.

**Figure 4 nutrients-13-02917-f004:**
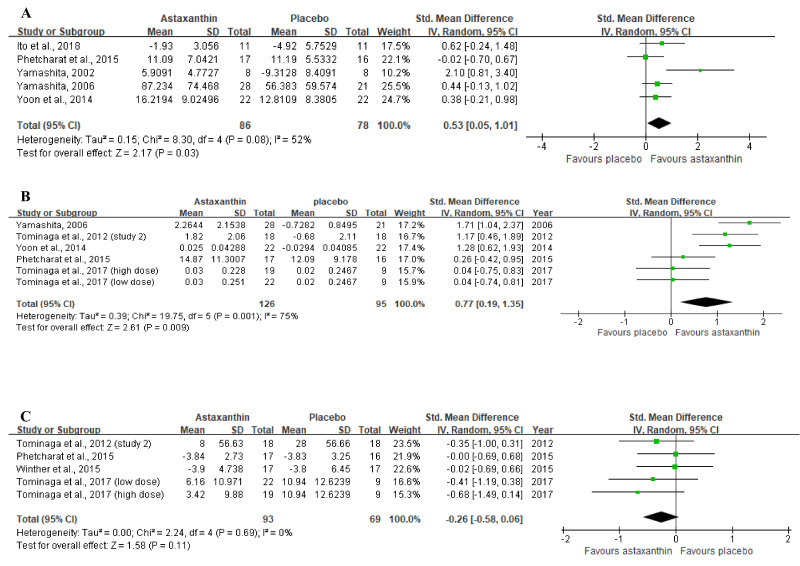
Forest plots summarizing the impact of ASX oral supplementation on (**A**) the increase in moisture content (µS), and (**B**) the increase in skin elasticity (%) using a random-effects model, as well as (**C**) the reduction in wrinkle depth (µm) using a fixed-effects model. Pooled summary data are presented as mean differences compared to control. The area of each green symbol is proportional to the weight of the study. Squares represent the standard mean differences, bars represent the 95% CI, and diamonds show the pooled effect.

**Figure 5 nutrients-13-02917-f005:**
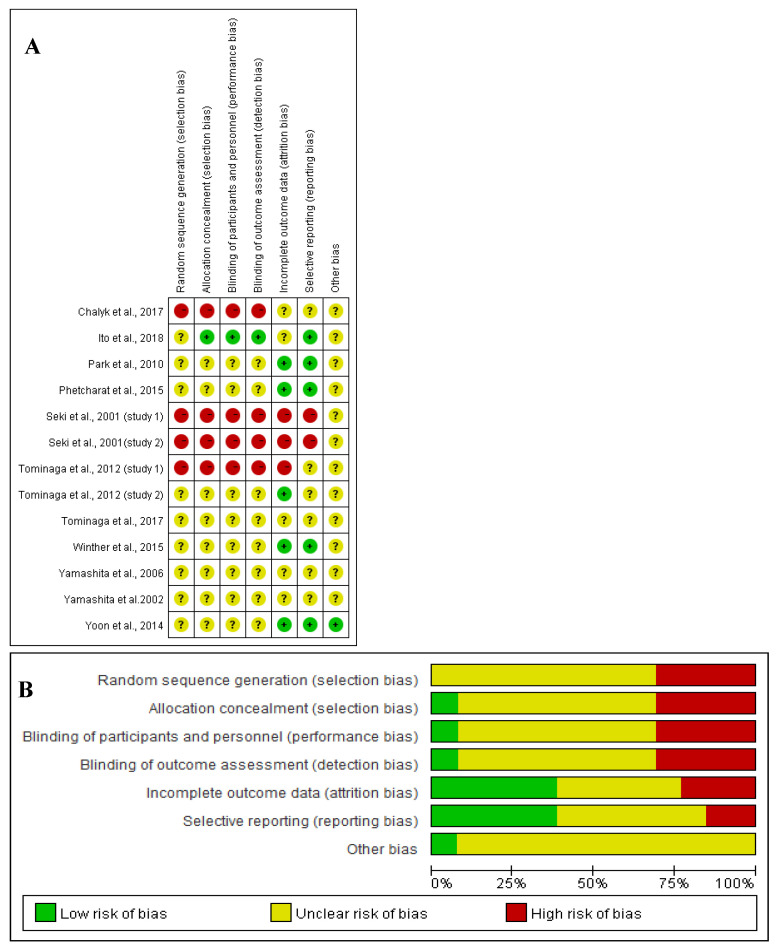
(**A**) Graph of RoB for each study according to the seven domains defined by Cochrane Collation’s tool. (**B**) Review of each RoB item presented as percentages across all included studies according to the judgments of the author.

**Table 1 nutrients-13-02917-t001:** Clinical trials investigating the effects of ASX on skin parameters (arranged by publication years).

Author, Year	Study Design	Study Population			Intervention				Control			Conditions of the Test Room (°C R.T.) (% R.H.)	Seasons during Study Periods and UV Applications	Study Duration (Weeks)	Outcome		
		Age, Years	Country	Health Condition	Participants Completed (*n*), Gender (W/M)	ASX Dosage (mg/d)	ASX Contents	Form (T/O)	Participants Completed (*n*), Gender (W/M)	Content	Dosage (mg/d)				Meaurement Instrument	Parameter (Measuring Sites)	Unit
Randomised Controlled Study																	
Yamashita, 2002 [28]	Randomised, placebo-controlled, double-blind study	Mean age of 40 years	Japan	Healthy with dry skin	8, W	2 40	*H. pluvialis* + canola oil tocotrienol	O	8, W	Canola oil	2	20 65	Winter	4	Electrical conductance-type SKICON-200 (IBS)). Transmission sebum-meter	Moisture content (ROC, RC) Sebum content (LC)	µS N/A
Yamashita, 2006 [34]	Randomised, placebo-controlled, single-blind study	Mean age of 47 years	US	Healthy (9 dry skin, 1 oily skin, 9 normal skin, and 30 combo skin)	28, W	4	*H. pluvialis* + canola oil	O	21, W	Canola oil	4	20 65	Winter	6	Dermal Phase Meter 9003 (NOVA meter) Cortex Technology Dermal Lab	Moisture content (LC) Skin elasticity (LOC)	µS %
Tominaga et al., 2012 (study 2) [29]	Randomised, placebo-controlled, double blind study	20–60	Japan	Healthy	18, M	6	*H. pluvialis* + canola oil	O	18, M	Canola oil	6	20 ± 2 45 ± 10	Autumn and Winter	6	ASA-CT1 ASA-03RXD image analysis ASA-M2 ASA-GP1 SEBU sheet around the nose	TEWL (LC) Wrinkle depth (LOC) Moisture content (LC) Skin elasticity (LOC) Sebum oil (LC)	g/h/m^2^µm µS % N/A
Yoon et al., 2014 [9]	Randomised, placebo-controlled, double blind study	41–60 (Mean 51.0 years)	South Korea	Healthy, skin wrinkles ≥ grade 2	22, W	2 3000	*H. pluvialis* + medium chain triglycerides + safflower oil as soft capsule Enzymatic hydrolysed fish collagen hydrolysate as tablet.	O & tablets	22, W	Medium chain triglycerides as capsule Hydrolysed casein as tablet	1000 3000	20–25 45–55	UV irradiation with two minimal erythema doses on the biopsies of buttock area (n = 6/group) for oxidative and inflammatory biomarkers assessments	12	Cutometer MPA580 Corneometer Tewameter Immunohistochemical staining Quantitative real-time PCR	Skin elasticity (cheek) Moisture content (cheek) TEWL (cheek)Thymine dimers and 8-OHdG Relative mRNA level of procollagen type I, fibrillin-1, MMP-1, and MMP-12	% µS g/h/m^2^ D D
Phetcharat et al., 2015 [32]	Randomised, double-blinded controlled study	35–65 (Mean 44.5 years)	Thailand	Healthy, well-defined crow’s-feet wrinkles or possibly other well-defined wrinkles on the face	16, W & M 17, W & M	3000 4	Rose hip powder (sourced from *R. canina* L.) containing seeds and shells + galactolipid *H. pluvialis* + rice bran oil	O	N/A	N/A	N/A	25 N/A	N/A	8	Visioscan^®^ VC 98 Corneometer^®^ CM 825 Cutometer^®^ MPA 580	Wrinkle depth (outer canthus) Moisture content (forehead) Skin elasticity (RC)	µm µS %
Winther et al., 2015 [35]	Randomised, single blind study	35–65	Thailand	Healthy	17, W & M 17, W & M	3000 4	Rose hip powder (sourced from *R. canina* L.) containing seeds and shells + galactolipid *H. pluvialis*	O	N/A	N/A	N/A	N/A	N/A	8	Skin Visioscan ^®^ VC 98	Wrinkle depth (outer canthus)	µm
Tominaga et al., 2017 [13]	Randomised, placebo-controlled, double blind study	35–60	Japan	Healthy, wrinkle grade on the lower and outer angle eyelids of 2.5 to 5.0	41, W (low dose: 22; high dose: 19)	6 or 12	*H. pluvialis* + canola oil	O	18, W	Canola oil	6 or 12	N/A	Summer, Autumn and Winter	16	PRIMOSLITE image analysis Skin hygrometer SKICON-200EX ASCT1 Cutometer MPA560 ELISA	Wrinkle depth (outer canthus) Moisture content (cheek) TEWL (cheek)Skin elasticity (cheek) IL-1α	µm µS g/h/m^2^ % pg/µg protein
Ito et al., 2018 [33]	Randomised, placebo-controlled, double blind study	30–56 (Mean 43.5 years)	Japan	Healthy, phototype II or III skin, second, third or fourth points basement MED in six-grade UV-irradiated area	11, W&M	4	*H. pluvialis* solvent-free capsule extracted by supercritical CO_2_ extraction technology	O	11, W&M	A filling agent	4	21 ± 1 50 ± 5	Autumn to Winter, 31.8, 36.5, 42.0, 48.3, 55.5 and 63.9 mJ/cm^2^ of UV-B before and after supplementation on the back skin	10	Visual assessment by an expert evaluator Corneometer^®^ VAPOSCAN AS-VT100RS	MED Moisture content (with 1.15 MED) TEWL (with 1.32 MED)	mJ/cm^2^µS g/m^2^
Open-label, prospective study																	
Seki et al., 2001 (study 1) [30]	Open-label, prospective study	20–50 (Mean 35 years)	Japan	Healthy	11, W	280	*H. pluvialis* + acetone + tri(capryl/capric acid) glycerol + cream base	T	N/A	N/A	N/A	21 55	N/A	3	Moisture Checker MY707S, Scalla Inc.	Moisture content (LOC)	%
Seki et al., 2001 (study 2) [30]	Open-label, prospective study	30–40 (Mean 33 years)	Japan	Healthy (40 es normal skin, 30 es dried skin, 30es mixed type)	3, W	140	*H. pluvialis* + acetone + tri(capryl/capric acid) glycerol + cream base	T	N/A	N/A	N/A	20 65	N/A	2	Skin Surface Image Analyzer and verted thickness (deepness)/length of wrinkles and size Moisture Checker MY707S, Scalla Inc.	Wrinkle depth (ROC) Moisture content (LOC, ROC, LC, RC)	D µS
Tominaga et al., 2012 (study 1) [29]	Open-label, prospective study	20–55	Japan	Healthy	30, W	6 94.18	*H. pluvialis* + canola oil *H. pluvialis*	O	N/A	N/A	N/A	20 ± 2 45 ± 10	Autumn and Winter	8	Skin surface photographs and replica images ASA-GP1 ASA-03RXD image analysis ASA-03RXD image analysis ASA-M2	Wrinkle depth (LOC) Elasticity (LOC) Mean depth of texture (LC) Total area of the corneocyte (LC) Moisture content (LC)	µm % µm µm^2^ µS
Additional study																	
Park et al., 2010 [27]	Randomised, placebo-controlled, double blind study	20.2–22.8 (Mean 21.5 years)	South Korea	Healthy	28, W	2 or 8	*H. pluvialis* (supercritical CO_2_ extract) containing small amounts (<15%) of mixed lutein, β-carotene and canthaxanthin	O	14, W	Lutein + β-carotene+ canthaxanthin	2 or 8	N/A	N/A	8	Commercially available ELISA kits Competitive ELISA kits	CRP 8-OHdG	µg/dL ng/mL
Chalyk et al., 2017 [31]	Open-label, prospective study	40–80 (Mean 61.45 years)	Russia	Healthy, 83.9% participants with clearly manifested ageing-related skin signs	31, W & M	4	*H. pluvialis*	O	N/A	N/A	N/A	N/A	N/A	4	Thiobarbituric acid reactive substances assay kit Cell^B imaging software Cell^B imaging software	Plasma MDA concentration Lipid droplet size (RSSC samples) Corneocyte desquamation (RSSC samples)	µMol/L µm number

## Data Availability

Not applicable.

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
