# Peer review of "Systematic Review and Meta-Analysis on the Effects of Astaxanthin on Human Skin Ageing"

_nutrients, 2021, doi:10.3390/nu13092917_

Round 1

Reviewer 1 Report

The paper title “Systematic review and meta-analysis on the effects of astaxan-2 thin on human skin ageing” represent an interesting work and in line with the scope of the journal.

However, some revision was necessary.

The aim of the study must be at the end of the introduction, but not in a separate item, remove 1.2 and adjust according.

Line 103: (Ng et al., 2020) is the reference [2].

Figure 3 is quite impossible to read, please made a new one in a readable form. And the explanation of this table must be improved, for example, in the Figure, Z is the test for overall effect, ins not defined but is defined for example “ .. overall effect size of 0.77 (95% CI = 0.19, 1.35) “. The authors explain how they are calculated but there aren´t explained in a clear way in the figure? All this part must be improved.

The Risk and Bias, must be better discussed, in fact the higher risk compromise the work.

Line 367 and 368, I think must be I2

Please revise the sentence from 504 to 506

Revise all references: ex: 40, and 41 appears after 46; I didn't find the references 42 and 43 in the manuscript.

“It is also desirable to apply supercritical CO2 extraction technology to generate a solvent-free and isolated form of ASX” this sentence do not make sense in this context and in this paper.

Hy the authors do not perform the Meta-regression and/or the Funnel plots? maybe could be possible extract more information.

Author Response

On behalf of my co-authors, we thank you for taking your time to review this manuscript. We are encouraged by your constructive comments and suggestions on our manuscript. We have accepted these comments with great respect and addressed all the issues raised. All the changes were labelled in red colour fonts in the revised manuscript. Attached is the response to reviewers’ comments, which we would like to submit for your kind consideration.

Reviewer 2 Report

The manuscript by Zhou and coworkers is a review aiming to highlight the inconsistencies of a number of studies performed so far on the anti-ageing properties of Astaxanthin in humans. In particular, the limits for a reliable interpretation of results are due to different endpoints of the studies, number of subjects investigated, different formulations used, lack of appropriate control and even possible Author’s conflict of interest.

The inconsistency of studies using components (not in pure/single formulations) to be added to food supplements or to cosmetics are not infrequent and therefore these inconsistencies are not surprising

The literature on the subject is not very large and it was appropriately cited and commented.

Author Response

We thank the reviewer for the time and effort in reviewing our manuscript.

Reviewer 3 Report

The review by Zhou et al. on Clinical Studies examined Astaxanthin in human skin aging in a detailed report with worthy information. The following comments are to assist in improving the review.

  1. Line 48-49 “reversing skin aging” is not doable; possibility state “slowing down skin aging.”
  2. Lines 60-76 presentation is good but to assist the reader please generate a new figure showing UVC, UVB and UVA to go along with the text.

See Wilkinson and Hardman, Mech. Aging Dev. 2021, 111513 and/or Antioxidants, 2021, 10(4), 578, doi:org/10.3390/antiox10040578 as examples.

  1. Table 1 is very difficult to read, increase the font size to at least 12.
  2. Figure 3 is also difficult to read, increase font size to 12, also for C the Std Mean Diff V, Random 95 % CI is reversed, please match to A and B to be consistent.
  3. Figure 4. Risk is unclear please make a clear summary statement in the figure legend and text to assist the reader in understanding this data.
  4. Section 4.3 please include the decline in estrogen levels (in women) as a major cause in declining skin health see Lephart & Naftolin, 2021, vol. 11, 53-69, Dermatology and Therapy as a new reference.
  5. The text is very detailed, please made clear summary statements in the abstract and conclusion so the reader is certain what the impact astaxanthin has on improving skin health.

Minor:

See lines, 316, 317, 318, 338, 342, 344, 345 and 537 for typos

Author Response

On behalf of my co-authors, we thank you for taking the time to review this manuscript. We are encouraged by the editors and reviewers for their constructive comments and scientific suggestions on our manuscript. We have revised the manuscript according to these comments and presented our rebuttal with great respect. All the alternations were labelled in red colour in our revised manuscript. Attached please find the response to your comments, which we would like to submit for your kind consideration.

This manuscript is a resubmission of an earlier submission. The following is a list of the peer review reports and author responses from that submission.